# Accuracy of Algorithms Predicting Accessory Pathway Localization in Pediatric Patients with Wolff-Parkinson-White Syndrome

**DOI:** 10.3390/children9121962

**Published:** 2022-12-14

**Authors:** Stefan Kurath-Koller, Martin Manninger, Nathalie Öffl, Martin Köstenberger, Hannes Sallmon, Joachim Will, Daniel Scherr

**Affiliations:** 1Division of Pediatric Cardiology, Department of Pediatrics, Medical University Graz, 8036 Graz, Austria; 2Division of Cardiology, Department of Medicine, Medical University of Graz, 8036 Graz, Austria; 3Department of Pediatric Cardiology, Charité—Medical University Berlin, 10117 Berlin, Germany

**Keywords:** WPW syndrome, pathway localization, electrophysiology, ablation, ECG

## Abstract

We aimed to assess the accuracy of determining accessory pathway (AP) localization from 12 lead ECG tracings by applying 12 different algorithms in pediatric patients diagnosed with Wolff-Parkinson-White syndrome. We compared algorithm accuracy in electrophysiologic study ECG tracings with full preexcitation and resting ECG tracings. The assessing pediatric cardiologists were blinded regarding EP study results on AP localization. For exact AP location, the algorithms published by D’Avila et al. and Boersma et al. yielded the highest accuracy (58%). Distinguishing laterality, the median accuracy for predicting left or right-sided APs was 74%, while for septal APs it was 68%. We conclude that algorithms predicting AP location in pediatric patients with Wolff-Parkinson-White syndrome show low accuracy for exact AP localization. For laterality, however, accuracy was significantly higher.

## 1. Introduction

Wolff-Parkinson-White (WPW) syndrome represents the most common tachycardia substrate in pediatric patients and is subject to electrophysiology study (EPS) and ablation therapy [1,2]. To enhance the efficiency of EPS by predicting accessory pathway (AP) location using the surface ECG, different algorithms have been developed [3,4,5,6,7,8,9,10,11,12]. However, except for one (Boersma et al. [11]) algorithm, the majority were developed and approved exclusively for adult patients. Due to the maturation of the conduction system and heart throughout childhood [13], one would expect the accuracy of the abovementioned algorithms to vary with age, being less accurate in young patients. This has been shown for some of the published algorithms, which yield a predictive accuracy for exact localization of about 50% [14,15]. As ventricular preexcitation is a major discriminating factor among these algorithms, ECG tracings with full preexcitation resulting from atrial pacing during EPS should yield the highest accuracy. The aim of this study was to assess the predictive accuracy of 12 published algorithms [3,4,5,6,7,8,9,10,11,12] for accessory pathway localization in pediatric WPW patients by comparing 12 lead resting ECG tracings with ECG tracings showing full ventricular preexcitation.

## 2. Materials and Methods

We conducted a retrospective single-blinded dual-center cohort study at the Division of Pediatric Cardiology, Medical University Graz, Austria, and the Department of Pediatric Cardiology, Charité University Hospital Berlin, Germany. Patient files were reviewed from 2016 to 2020. Patients aged 0–18 years, diagnosed with WPW syndrome with a manifest ventricular preexcitation pattern in 12 lead resting ECG tracings, were identified and included if AP localization was confirmed by EPS. The localization of AP from EPS was considered the gold standard. Exclusion criteria comprised presence of multiple APs, congenital heart disease (CHD), or lack of adequate ECG quality for assessment (e.g., not all leads documented and multiple artifacts), unsuccessful ablation, recurrence of preexcitation, and no antegrade conduction via the accessory pathway.

ECG tracings: We used accessory pathway determination algorithms that have been published in the literature (Taguchi et al. [12], Iturralde et al. [4], DÀvila et al. [7], Arruda et al. [3], Fitzpatrick et al. [6], Milstein et al. [8], Chiang et al. [5], Lindsay et al. [10], Xie et al. [9], Boersma et al. [11], Li et al. [16], and Baek et al. [17]). To assure reliable results, the two assessing pediatric cardiologists (both trained in electrophysiology for a minimum of 2 years) were blinded to the EPS results regarding AP localization. The exact AP localization as well as laterality (i.e., right, left, or septal AP) were assessed by applying each algorithm to 12 lead resting ECG tracings and fully preexcited ECG tracings during atrial pacing.

EP Study: At the center in Graz, EPS was conducted using the CARTO^®^ 3 System Version 6 (Biosense Webster Inc., 31 Technology Drive, Suite 200, Irvine, CA 92618, USA). At the center in Berlin, the EnSite Precision™ Cardiac Mapping System (Abbott Laboratories, 100 Abbott Park Road, Abbott Park, IL 60064, USA) was used. Full preexcitation was achieved using atrial pacing at incremental cycle lengths (programmed stimulation from the high right atrium with incrementing S2 until the AV node’s effective refractory period).

Accessory pathway localization: The 12 algorithms used within this study use different nomenclature regarding exact AP location. A total of 32 different terms are used. We sought to translate algorithm nomenclature to fit the consensus statement from the Cardiac Nomenclature Study Group, the Working Group of Arrhythmias, the European Society of Cardiology, and the Task Force on Cardiac Nomenclature from NASPE [18], where 13 terms are proposed to describe AP localization (Table 1).

SPSS 27 (IBM Corp., Armonk, NY, USA) was used for data analysis, and *p* < 0.05 was considered significant. Missing data were not imputed. The agreement of the different algorithms with the gold standard (AP localization from EPS) was calculated using Cohen’s Kappa. Ninety-five percent confidence intervals were calculated using Bootstrapping methods. The used bootstrapping algorithm draws 1000 samples using simple Bootstrapping resampling. When the procedure following the Bootstrapping algorithm is run, the pooling algorithm produces 95% confidence intervals using the percentile method. The percentage of agreement is also reported. For laterality, statistical accuracy was calculated as (true positive/true negative)/total number of probands and given as a percent (%).

## 3. Results

Overall, 64 patients (28 female, 36 male) with a mean age at the time of EPS of 15 years (ranging from 6 to 17 years) were included. There was no statistically significant difference between the two centers with regard to patient age, gender, or number of right, left, or septal APs. Overall, 128 ECG tracings (64 resting ECG tracings and 64 ECG tracings with full ventricular preexcitation during EPS) were included and analyzed. AP localization sites from EPS were left lateral (LL, n = 26), left posteroseptal (RPS, n = 13), right posterolateral (RPL, n = 7), left posteroseptal (LPS, n = 5), right septal/midseptal (RS, n = 3), left posterolateral (LPL, n = 3), right lateral (RL, n = 2), right anterolateral (RAL, n = 2), right anterior (RA, n = 2), right anteroseptal (RAS, n = 1), and left epicardial (n = 1). No statistically significant differences were found when comparing the cohorts of the two participating centers regarding time of ECG acquisition and EPS, accuracy rates for exact AP localization, or laterality. The mean time from 12 lead resting ECG tracing acquisition to EPS was 34 days (range 2–73 days). For 12 lead resting ECG tracings, the algorithms published by D’Avila, Boersma, and Xie [7,9,11] were found most accurate to determine exact AP location (58%, 54%, and 54%, respectively). For laterality, the highest accuracy was found for the prediction of right-sided APs (from 42 to 78%; median 74%). For left-sided APs the median accuracy was 73% (from 53 to 80%), and for septal APs, the median accuracy was 68% (from 48 to 78%).

From ECG tracings with full preexcitation during EPS the algorithm published by Boersma [11] yielded highest accuracy rates for exact AP localization (55%). For laterality, highest accuracy was found for prediction of right-sided APs (from 52 to 78%; median 73%). For left sided APs median accuracy was 73% (from 58 to 83%), and for septal APs median accuracy was 65% (from 50 to 75%).

No statistically significant differences were found between results from 12 lead resting ECG tracings vs. ECG tracings with full preexcitation during EPS regarding laterality (right-sided APs, *p* = 0.502; left-sided APs, *p* = 0.502; and septal APs, *p* = 0.509).

When the results from the two participating centers were compared, no statistically significant differences in accuracy for predicting AP localization were found. Additionally, no statistically significant differences were found when comparing younger children (aged from 6 to 12 years /BSA from 0.9 to 1.3 m^2^) with adolescents (age > 12 years; BSA > 1.3 m^2^) regarding the accuracy of the studied algorithms.

Figure 1 shows prediction accuracy for laterality from 12 lead resting ECG tracings and fully a preexcited ECG during EPS.

## 4. Discussion

Recent guidelines incorporate determination of accessory pathway location from 12 lead resting ECG tracings prior to EPS and ablation therapy. Their use may reduce procedure duration and associated radiation burden for both patients and physicians [19,20]. Furthermore, pre-ablation determination of AP location may lead to higher success rates and lower costs for health care systems. 

In pediatric patients, it has been shown that left-sided accessory pathways are associated with lower complication rates from ablation therapy, while right posteroseptal APs are associated with higher recurrence rates. Prediction of either of these two pathway locations might influence decision making regarding ablation therapy. 

However, we found that the accuracy of the currently available algorithms to determine accessory pathway location from 12 lead resting ECG tracings in pediatric patients is rather low. The algorithms by Boersma, D’Avila, and Xie [7,9,11] showed the best performance for exact AP localization in our pediatric patient cohort. Nevertheless, determination of the exact AP location remains somewhat imprecise. For laterality, all algorithms showed higher accuracy. ECG quality should be expected to be superior in pediatric patients with respect to body fat and fewer comorbidities. On the other hand, the younger the children, the more likely it is that movement artifacts will impair tracing quality. All patients included in this study had clear recordings, displaying sharp ECG signals without artifacts.

Since many algorithms are based on the ventricular preexcitation pattern, one would expect ECG tracings with full preexcitation, e.g., during rapid atrial pacing, to yield higher accuracy rates. Interestingly, our data suggest that this is not the case. We found that assessment of fully preexcited ECG tracings was not superior to 12 lead resting ECG tracings. This seems to be, at least in part, due to minor changes in ECG key features for AP localization (e.g., R/S ratio in V1 and V2, preexcitation pattern in lead III, AVF, and V1, among others). Furthermore, in 12 lead resting ECG tracings, preexcitation patterns did not significantly differ from full preexcitation patterns during rapid pacing in EPS (Figure 2). However, including larger numbers of ECG tracings for analysis may alter these findings.

Full preexcitation pattern derived from atrial pacing during EPS may be of greater value in cases of sole discrete preexcitation pattern in 12 lead resting ECG tracings. This, however, needs to be addressed in a larger follow-up study. After all, the value of an algorithm derives from the information it provides to the informed consent process and the evaluation of risk versus benefit of electrophysiological intervention.

Considering the development from childhood to young adults, our findings of rather low accuracy are not surprising. Except for the algorithm by Boersma et al. [11], none were derived from, or designed for, pediatric patients. Taking the growth and developmental changes of the heart and cardiac conduction system into account might improve accuracy of future ECG algorithms. Our results call for an intrinsic algorithm derived from, and designed for, pediatric patients. Our cohort, however, includes too few ECG tracings to attempt construction of such a new algorithm. Instead, a multicentric approach incorporating collaborative networks and including some thousand pediatric ECG tracings with manifest preexcitation is warranted. Furthermore, significant improvements could be made by using deep learning and artificial intelligence [21] for pattern determination and algorithm construction. Nishimori et al. [21] implemented a convolutional neural network model using chest X-ray and ECG data, achieving accuracy rates of up to 80%. Escobar et al. [22] published a machine learning algorithm able to identify septal APs with accuracy rates up to 97% in children with WPW-syndrome. However, deep learning-based algorithms require large amounts of sample data to be trained and develop accurate results. Global cooperation networks on this research topic could provide this “big data” and enhance research output. Despite promising results and high expectancies in the future, computer-based algorithms remain mostly unavailable to clinicians in daily practice

Of further notice, terminology regarding AP localization can become quite confusing when applying several different algorithms. Although algorithms have been proposed throughout different eras (from 1987 to 2020), those published after the release of the Consensus Statement from the Cardiac Nomenclature Study Group [18] in 1999 still lack uniformity of nomenclature. Eventually, after almost 25 years it might be necessary to reconcile AP localization nomenclature by unification. This may also enhance accuracy of results from comparative studies on AP localization algorithms for the future.

## 5. Conclusions

Accurate prediction of exact accessory pathway localization in pediatric WPW syndrome may be achieved in up to 58%. Determination of laterality (i.e., right, left, or septal pathway location) generally works better than prediction of exact localization and was most accurate for right sided APs (accuracy rates up to 82%). Rapid atrial pacing during EPS does not significantly enhance accuracy of predictive algorithms.

## 6. Limitations

The impact of our results is limited due to the relatively low number of tracings analyzed for each AP location. Furthermore, the study design was retrospective, and we did not assess inter- or intra-observer variability. While helpful, ECG tracings are easily altered by movement artifacts or electromagnetic interferences, depend on, and are unlikely to ever be 100% effective in predicting the precise ventricular insertion of an accessory pathway.

## Figures and Tables

**Figure 1 children-09-01962-f001:**
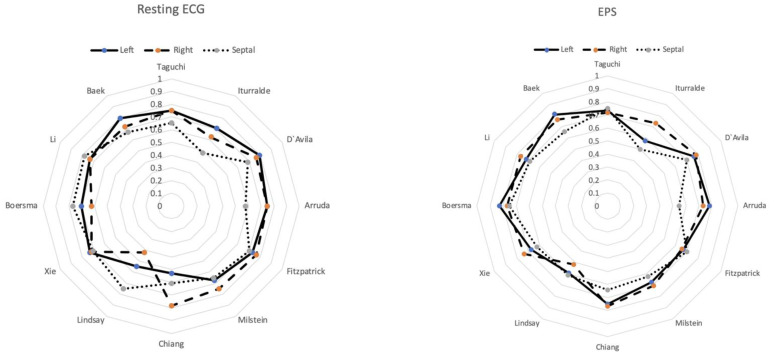
Prediction accuracy for laterality from 12 lead resting ECG tracings and ECG tracings with full ventricular preexcitation during EPS.

**Figure 2 children-09-01962-f002:**
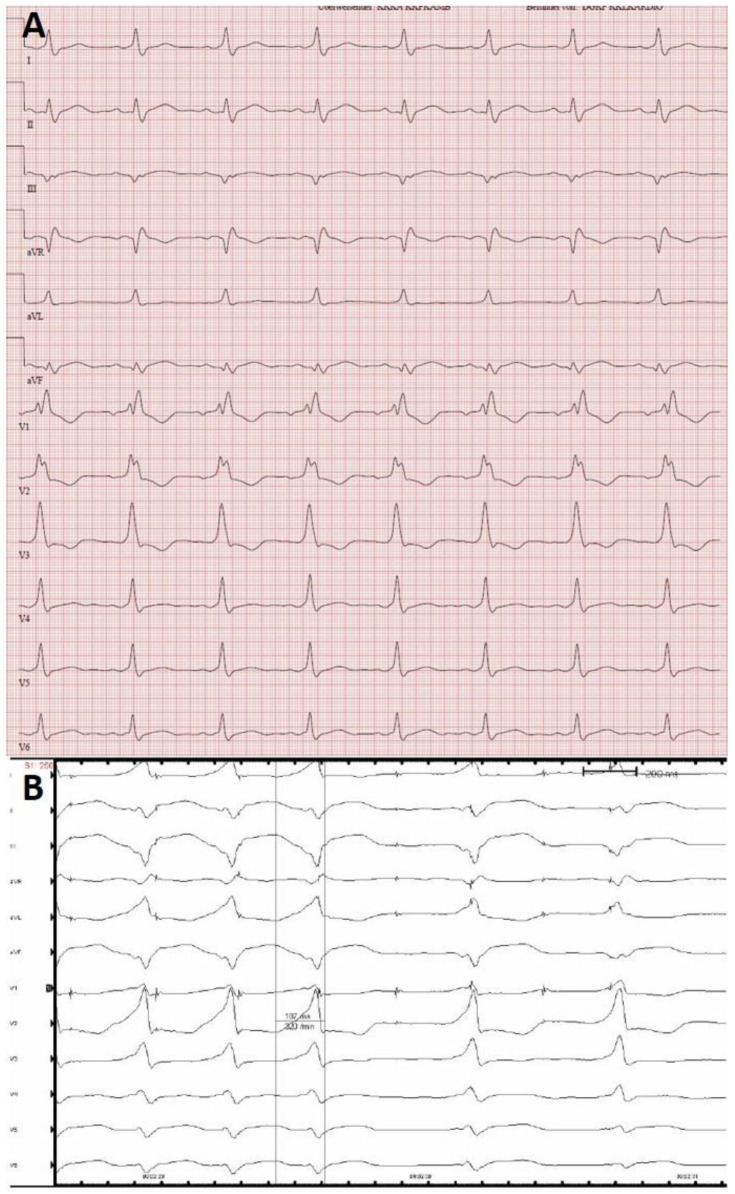
ECG tracings from a patient with a left posterolateral AP. (**A**) 12 lead resting ECG tracing with manifest preexcitation (50 mm/s). (**B**) 12 lead ECG tracing during atrial pacing in EPS shows a full ventricular preexcitation pattern (100 mm/s).

**Table 1 children-09-01962-t001:** Description of accessory pathway locations from the consensus statement from the Cardiac Nomenclature Study Group, the Working Group of Arrhythmias, the European Society of Cardiology, and the Task Force on Cardiac Nomenclature from NASPE [18] and the studied algorithms.

Laterality	Attitudinally Correct	Algorithms Terms
Right	Superior	Right anterior [3,5,12,16] Right antero-lateral [6,16] Anteroseptal [7,8,11] Right antero-septal [9] Right free wall [10,17]
	Supero-anterior	Right anterior [4,12] Right antero-lateral [3,5,6,16] Right lateral [7,8,16] Right free wall [10,17] Right lateral [11]
	Anterior	Right lateral [3,5,7,8,11,12] Right anterior [4] Right antero-lateral [6] Right postero-lateral [6] Right free wall [10,17]
	Infero-anterior	Right postero-lateral [3,5,6,12,16] Right inferior [4] Right lateral [7,8,11] Right free wall [10,17]
	Inferior	Right posterior [3,5,12,16] Right inferior paraseptal [4] Right paraseptal [7] Right lateral [8] Right free wall [10,17] Right postero-septal [11]
Left	Superior	Left antero-lateral [5,9,16] Left lateral [7,8,11] Left lateral free wall [10] Left free wall [17]
	Supero-posterior	Left anterior [12] Left antero-lateral [3,5,6,9,16] Left antero-superior [4] Left lateral [7,8,11] Left lateral free wall [10] Left free wall [17]
	Posterior	Left lateral [3,5,7,8,11,12,16] Left postero-lateral [4,9] Left antero-lateral [6] Left lateral free wall [10] Left posterior [16] Left free wall [17]
	Infero-posterior	Left postero-lateral [3,5,6,9,12,16] Left inferior [4] Left posterior [7] Left lateral [8,11] Left lateral free wall [10] Left free wall [17]
	Inferior	Left posterior [3,5,9,12] Left inferior paraseptal [4] Left paraseptal [7] Left lateral [8] Left posterior free wall [10] Left postero-septal [11] Left free wall [17]
Septal	Superoparaseptal	Right antero-septal [5,12,16] Antero-septal [3] Right antero-paraseptal [3,10] RAS paraseptal [4]
	Inferoparaseptal	Paraseptal [12] PS MA/TA [3] Right postero-septal [5,6,9,16,17] Left infero-paraseptal [5] Left postero-septal [6,9,16,17] Postero-septal [7,8,10,11]
	Septal	Midseptal [5,7,9,11] Right antero-septal [6,17] Right midseptal [16] Right midseptal [17]

Leg: RAS = right anterior superior; PS = posteroseptal; MA = mitral annulus; TA = tricuspid annulus.

## Data Availability

Underlying data may be provided by the corresponding author upon reasonable request.

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
