# Peer review of "Accuracy of Algorithms Predicting Accessory Pathway Localization in Pediatric Patients with Wolff-Parkinson-White Syndrome"

_children, 2022, doi:10.3390/children9121962_

Round 1

Reviewer 1 Report

This is an overall clearly and concisely written manuscript that touches on a high-yield topic in electrophysiology. It is clear that the results from this study can be embraced and applied by electrophysiologists in different settings. Thought it is a fairly small sample size, it is a worthy contribution to the field. A few comments and suggestions:

- Though not severely impacting the content of the manuscript, it would have been nice to see some more details on the characteristics of the population studied. Perhaps a full table 1 isn't necessary in this case but I would have wanted to know how it was determined that the populations were similar between the two hospitals. Was it solely based on age and gender? It would be nice to know if any of the children studied had congenital heart disease, for example.

- Similarly, it would be nice to know more about the cardiologists involved in the study. Were they general pediatric cardiologists or electrophysiologists? How many years of experience do they have? Did this vary between the 2 centers? Was every algorithm used by every cardiologist when assessing each ECG?

- The first few sentences of the Discussion seem to have been left there by error.

Author Response

Thank you for your comments, helping to improve our manuscript. We have thoroughly worked the manuscript and incorporated your suggestions and comments.

  • the groups matched by age and gender. We have included a sentence with this regard on page 3, line 107.
  • CHD was one of the exclusion criteria (see text page 2, line 63), so none of the patients had CHD.
  • information regarding the assessing cardiologists was added on page 2, line 71. Both were trained electrophysiologists.

We think the manuscript clearly improved and hope to have satisfied your demands.

sincerely

Stefan

Reviewer 2 Report

  This study offers insight to a particular syndrome – Wolff-Parkinson-White – and enables researchers to have a better understanding of the algorithms predicting AP localization in pediatric Wolff-Parkinson-White

Abstract: D’Avila et al. and Boersman et al. please check citations (18-19)

The Materials and Methods are clear and well justified for the presented study.

Discussion: the tone with which this section begins could be improved. (129-130)

The findings illuminate understanding within a specific context and are described and analyzed with clarity.   I found the article to be well-written and fluent. The findings are of interest and make an important contribution to the Wolff-Parkinson-White studies.

Author Response

Thank you for your comments, helping to improve our manuscript. We have thoroughly worked the manuscript and incorporated your suggestions and comments.

We think the manuscript clearly improved and hope to have satisfied your demands.

sincerely

Stefan